# Understanding PPARγ and Its Agonists on Trophoblast Differentiation and Invasion: Potential Therapeutic Targets for Gestational Diabetes Mellitus and Preeclampsia

**DOI:** 10.3390/nu15112459

**Published:** 2023-05-25

**Authors:** Yushu Qin, Donalyn Bily, Makayla Aguirre, Ke Zhang, Linglin Xie

**Affiliations:** 1Department of Nutrition, Texas A&M University, College Station, TX 77843, USA; yushu@tamu.edu (Y.Q.); donalyn.bily@tamu.edu (D.B.); mak1019aguire@gmail.com (M.A.); kzhang@tamu.edu (K.Z.); 2Department of Biology, Texas A&M University, College Station, TX 77843, USA; 3Institute of Biosciences and Technology, Texas A&M University, Houston, TX 77030, USA

**Keywords:** PPARγ, rosiglitazone, trophoblast, placenta, gestational diabetes mellitus, preeclampsia, pregnancy

## Abstract

The increasing incidence of pregnancy complications, particularly gestational diabetes mellitus (GDM) and preeclampsia (PE), is a cause for concern, as they can result in serious health consequences for both mothers and infants. The pathogenesis of these complications is still not fully understood, although it is known that the pathologic placenta plays a crucial role. Studies have shown that PPARγ, a transcription factor involved in glucose and lipid metabolism, may have a critical role in the etiology of these complications. While PPARγ agonists are FDA-approved drugs for Type 2 Diabetes Mellitus, their safety during pregnancy is not yet established. Nevertheless, there is growing evidence for the therapeutic potential of PPARγ in the treatment of PE using mouse models and in cell cultures. This review aims to summarize the current understanding of the mechanism of PPARγ in placental pathophysiology and to explore the possibility of using PPARγ ligands as a treatment option for pregnancy complications. Overall, this topic is of great significance for improving maternal and fetal health outcomes and warrants further investigation.

## 1. Introduction

Pregnancy can lead to complications that pose serious risks to both the mother and infant during pregnancy, labor, and postpartum. These complications typically arise from conditions unique to pregnancy. Alarmingly, there has been a 16.4% increase in the incidence of pregnancy complications between 2014 and 2018, with gestational diabetes mellitus (GDM) increasing by 16.6% and preeclampsia (PE) increasing by 19% [1]. Although it is recognized that the pathologic placenta is the root cause of many pregnancy complications, the exact mechanism is not yet fully understood. Recent clinical studies have suggested that genetic analysis, such as Peroxisome proliferator-activated receptor-γ (PPARγ) as a transcription factor, can offer a novel approach to diagnosis and prediction [2]. PPARγ is crucial for metabolism homeostasis, adipocyte differentiation, and the immune system. Research has revealed significant associations between certain PPARγ gene variations and PE, underscoring the importance of PPARγ in the development of this condition. Notably, PE is more prevalent in women with hyperglycemia, a well-known risk factor [3,4,5]. Women with diabetes are at least twice as likely to develop PE, with around 50% of diabetic pregnancies experiencing hypertensive disorders of pregnancy (HDP), particularly those with pre-existing diabetes and poor glycemic control [6,7,8,9]. Considering the therapeutic potential of PPARγ agonists, which are FDA-approved for Type 2 Diabetes Mellitus, it becomes evident that these agents hold promise for preeclampsia treatment, particularly in patients with risk factors such as hyperglycemia. Therefore, this review aims to provide a comprehensive overview of the current research on the mechanisms of PPARγ and the effects of PPARγ agonists on placenta pathophysiology. Such understanding paves the way for precision medicine strategies to prevent or mitigate the risk of PE, taking into account individual factors such as race, genetics, and maternal risk factors.

To examine the role of PPARγ in GDM and PE, with a particular focus on placental pathophysiology, this review conducted a comprehensive search using the PubMed database. Only original research and scientific abstracts published between 2005 and 2022 that investigated the role of PPARγ in GDM and PE were included. The search terms used were “peroxisome proliferator-activated receptor-gamma”, “PPARγ”, “PPAR gamma”, “gestational diabetes mellitus” and “preeclampsia”. Studies involving other PPARs and other pregnancy complications such as infertility, hypertensive pregnancy, and polycystic ovarian syndrome were excluded (Figure 1).

## 2. Peroxisome Proliferator-Activated Receptor-γ

The Peroxisome Proliferator-Activated Receptor-γ (PPARγ) is a PPAR subfamily member consisting of two isoforms. PPARγ1 is encoded by mRNA PPARγ1, PPARγ3, and PPARγ4, while PPARγ2 is translated from mRNA PPARγ2 [10]. PPARγ1 is broadly expressed in various tissues including adipose tissue, the liver, colon, heart, epithelial cells, and skeletal muscle, and is also found in immune cells such as monocytes/macrophages, dendritic cells, and T lymphocytes [11]. On the other hand, PPARγ2 contains 28 additional amino acids and is primarily found in adipose tissue. Both isoforms are highly expressed in reproductive organs such as the placenta, testis, and ovary [12].

PPARγ is a ligand-dependent transcription factor, meaning it can be regulated by agonists and antagonists. It acts as a sensor for different fatty acid types, also known as a lipid sensor. In addition to endogenous ligands, synthetic ligands are widely used in clinical practice and in vitro studies to modulate PPARγ [13,14]. A summary of reported PPARγ ligands is provided in Table 1.

PPARγ, in addition to its well-established roles in lipid metabolism and adipocyte differentiation, has also been shown to be essential in regulating insulin resistance, glucose metabolism, immunity, as well as cell biology, including cell differentiation [45,46,47]. The TZD family comprises FDA-approved drugs used for treating Type 2 Diabetes Mellitus [13]. Beyond its function in immunology and maintaining energy homeostasis, PPARγ is also indispensable for the early development of the conceptus as early as E10. Its critical role in development seems to be particularly important in the placenta [48]. This review primarily focuses on the function of PPARγ in trophoblast differentiation and invasion, as well as its relationship with pregnancy complications, including GDM and PE.

## 3. PPARγ Functions in the Placenta and Trophoblasts

PPARγ is highly expressed in human placentas, particularly in syncytiotrophoblasts, cytotrophoblasts, and extravillous trophoblasts (EVTs) [49,50]. Its expression in the placenta is associated with infant birth weight. Placentas from small-for-gestational-age (SGA) infants were found to have lower expression of PPARγ, whereas placentas from average-for-gestational-age and large-for-gestational-age infants showed a nearly 2-fold higher expression of PPARγ compared with that from SGA infants [51]. These findings suggest that PPARγ may play a role in regulating fetal growth and development in the placenta.

Recent in vitro studies have shown that PPARγ is associated with trophoblast migration and invasion, although its exact role in these processes appears to be paradoxical. Some studies have reported that PPARγ inhibits trophoblast invasion in human primary cultures of EVTs [52,53,54]. One proposed mechanism is through the repression of pregnancy-associated plasma protein A, which reduces insulin-like growth factor (IGF) availability and limits trophoblast invasion [55,56]. Additionally, heme oxygenase-1 (HO-1) has been reported to negatively regulate trophoblast motility through the up-regulation of PPARγ [57]. Furthermore, PPARγ has been shown to inhibit trophoblast migration through its interaction with endocrine gland-derived vascular endothelial growth factor (EG-VEGF), a placental angiogenic factor [58]. Some studies have also reported that rosiglitazone, a PPARγ agonist, blocked lipopolysaccharide (LPS)-induced invasion in human first-trimester trophoblast cell lines [59].

However, more recent studies have suggested that PPARγ may promote trophoblast migration. Activated PPARγ/RXRα heterodimer by IL-17 was found to promote proliferation, migration, and invasion in HTR8/SVneo, a trophoblast cell line [60]. Furthermore, pioglitazone, which increases PPARγ expression, was shown to stimulate EVT migration by promoting IGF signaling [56]. In addition, mutations on the ligand-binding domain of PPARγ have been found to significantly suppress migration in the primary villous cytotrophoblasts [61]. These findings suggest that the role of PPARγ in trophoblast migration and invasion may be complex, and further research is needed to fully understand its mechanisms and effects in these processes.

PPARγ has also been identified as a regulator of trophoblast differentiation. In the BeWo cell model, blocking PPARγ activity has been shown to induce cell proliferation but suppress the differentiation [62]. In human placenta explants, PPARγ/RXRα heterodimers have been found to promote cytotrophoblast differentiation into syncytiotrophoblasts [63], which is a key event in placental development. In PPARγ-deficient mouse placentas, diminished expression of several trophoblast differentiation markers, such as Tpbpα and Mash2, as well as the abnormal spatial expression of glial cell missing 1 (GCM1), a transcription factor important for syncytiotrophoblast differentiation, were observed [64]. In addition, oral administration of troglitazone, a PPARγ agonist, was found to enhance cytotrophoblast differentiation into syncytiotrophoblasts [65]. PPARγ also promotes the differentiation of syncytiotrophoblasts, but not trophoblast giant cells (TGCs) in the mouse labyrinth, which is the region of the placenta where nutrient exchange occurs [66]. On the other hand, rosiglitazone, another PPARγ agonist, has been reported to reduce TGC differentiation while inducing GCM1 expression [67], suggesting that the role of PPARγ in trophoblast differentiation may be complex and dependent on the specific cell type. Further research is needed to fully elucidate the mechanisms and effects of PPARγ in trophoblast differentiation.

## 4. Genome-Wide Association Studies (GWAS) Suggested That PPARγ Is Associated with Preeclampsia and Gestational Diabetes Mellitus

PPARγ single nucleotide polymorphisms (SNPs) are associated with increased susceptibility to pregnancy-related diseases, including GDM and PE. The rs201018 and C1431T variants of PPARγ have been reported to be significantly associated with susceptibility to PE in different populations [54,55], with the rs201018 polymorphism showing a correlation with the incidence of PE in the Chinese population [55], and the C1431T polymorphism is associated with PE occurrence in the French population [54].

In addition to PE, PPARγ SNPs have also been associated with the incidence of GDM. The Pro12Ala polymorphism of PPARγ is one of the dominant variants associated with GDM susceptibility, as reported in several meta-analyses [56,57,58]. However, there are conflicting findings regarding the role of Pro12Ala in GDM, with some studies suggesting a protective role against GDM in certain populations, such as the Filipino population, while others suggest that it may exacerbate insulin resistance by elevating serum resistin levels [59]. Besides Pro12Ala, the rs1801282 variant of PPARγ is associated with increased GDM incidence in Russian [60] and Asian [61] populations, but not in the Brazilian population [62]. Recent studies have also suggested that the PPARΓ (rs1801282) variant may be a significant risk factor for the development of PE in women with GDM in the Russian population [63]. These findings highlight the potential role of PPARγ SNPs in modulating the risk of pregnancy-related diseases, although further research is needed to fully understand the underlying mechanisms and implications of these genetic variants. Large epidemiological studies of the population considering different demographic information will be essential to establish a comprehensive understanding of PPARγ SNPs in modulating the risk of pregnancy-related diseases.

## 5. The Role of PPARγ in Preeclampsia

PE is a serious pregnancy complication that affects 5–7% of pregnancies worldwide [68] and is responsible for over 500,000 maternal and fetal deaths each year [69]. PE is considered the leading cause of maternal morbidity and mortality in the United States, accounting for 16% of maternal deaths [70]. Women diagnosed with PE are at increased risk of adverse outcomes not only for themselves, but also for their babies in the future.

PE can have serious consequences for both the mother and the baby. If the placenta is not implanted correctly, it can result in inadequate blood flow to the placenta, leading to fetal growth restriction or intrauterine growth restriction (IUGR) [71]. This can result in babies being born with low birth weight and other health complications. Additionally, women who have experienced PE have an increased risk of developing cardiovascular disease later in life, as well as their children [72].

Diagnosis of PE typically involves measuring blood pressure, with systolic blood pressure greater than 140 mmHg and diastolic blood pressure greater than 90 mmHg, along with proteinuria [69]. PE is usually diagnosed after 20 weeks of pregnancy, but it can occur in both early and late pregnancy. The development of PE is believed to involve two interconnected stages. The first stage is usually asymptomatic and is initiated by inadequate placental circulation due to abnormal placental implantation and/or reduced blood flow to the placenta. This can result in placental ischemia, or reduced blood flow to the placenta, due to aberrant failed remodeling of spiral arteries and impaired vascularization [73]. If placental perfusion remains compromised, it can progress to the second stage of PE, which is characterized by clinical manifestations such as high blood pressure and other symptoms [74]. Understanding the pathogenesis of PE is complex, and ongoing research is needed to further elucidate the underlying mechanisms and develop effective prevention and treatment strategies.

### 5.1. PAPRγ in the Pathogenesis of PE

The role of PPARγ in the pathogenesis of PE is still not fully understood, but studies have shown conflicting results regarding its expression levels in PE placentas. While some studies have reported increased expression of PPARγ in PE placentas [75,76], most studies have found decreased expression of PPARγ in PE placentas [76,77,78,79], while the expression of PPARγ increased in the blood serum of PE patients [80].

One mechanism that has been explored is the modulation of 11β-hydroxysteroid dehydrogenase type 2 (11β-HSD2), a gene that helps in the maintenance of cortisol levels in the placenta. PPARγ was found to positively correlate with 11β-HSD2 expression in PE placentas, and treatment with rosiglitazone, a PPARγ agonist, increased the expression of 11β-HSD2 in placental explants, while treatment with GW9662, a PPARγ antagonist, decreased the expression of 11β-HSD2. Interestingly, the effect of PPARγ ligands was blocked when specificity protein 1 (Sp-1) was knocked out [79]. Another transcription factor that has been implicated in the pathogenesis of PE is GCM1, which regulates trophoblast differentiation [81]. GCM1 expression is significantly downregulated in PE placentas, and depletion of GCM1 in normal placental explants elevated the secretion of soluble Fms-like tyrosine kinase 1 (sFlt-1), a marker for PE. Treatment with rosiglitazone increased GCM1 expression, while treatment with T0070907, a PPARγ antagonist, reduced GCM1 expression [78].

PPARγ has also been shown to participate in regulating sFlt-1 secretion through nuclear factor erythroid 2-related factor 2 (Nrf2) [82]. Procyanidin B2, a natural compound, reduced sFlt-1 secretion and restored the migration capacity of trophoblasts in placental explants from PE pregnancies by activating Nrf2, which bound to the promoter region of PPARγ and enhanced its transcriptional activity.

In addition, studies have shown that PPARγ and angiopoietin-like protein 4 (ANGPTL4), which is associated with fat metabolism and vessel formation, are reduced in PE placentas compared with control placentas [83]. Treatment with rosiglitazone upregulated the expression and secretion of ANGPTL4 in placental explants [83], suggesting a potential interaction between ANGPTL4 and PPARγ in the pathogenesis of PE.

PPARγ has been implicated in regulating epigenetic modifications in the placenta, specifically histone methylation and acetylation. Epigenetic modifications play a crucial role in regulating gene expression and can impact various cellular processes, including trophoblast invasion and migration [84]. The co-expression of PPARγ with histone markers such as H3K4me3 and H3K9ac is upregulated in preeclamptic placenta [85], suggesting a potential involvement of PPARγ in placental epigenetic regulation in the context of PE. Treatment with ciglitazone, a PPARγ agonist, has been shown to restore the levels of these histone modifications, while treatment with T0070907, a PPARγ antagonist, further induced an increased level of H3K4me3 and H3K9ac [85]. This suggests that PPARγ may play a role in modulating placental epigenetic modifications in PE, although the exact mechanisms and implications of these epigenetic changes are still not fully understood and require further research. One of the potential reasons is the variation in cell populations and epigenetic profiles of placentas among individuals [86,87]. This requires large sample sizes to comprehensively determine the effect of epigenetic-modified PPARγ.

Overall, the role of PPARγ in the pathogenesis of PE is complex and still not fully understood, with conflicting findings in different studies. Further research is needed to elucidate the exact mechanisms and potential therapeutic implications of PPARγ in PE.

### 5.2. Effect of PAPRγ Ligands on PE in Rodent Models

Although human placental tissue and explants can provide valuable information about the function of PPARγ in PE, ethical concerns make it impossible to determine the effect of its ligands in human studies. Therefore, rodent models, such as mice and rats, have been widely used to study the molecular mechanisms of PE. One commonly used rodent model is the reduced uterine perfusion pressure (RUPP) model in pregnant rats, which involves creating an abdominal incision on E14.5 of gestation to mimic abnormal uteroplacental blood flow [88]. RUPP rats exhibit characteristics of PE, including high blood pressure, impaired vasorelaxation, and elevated ACR (albumin-to-creatinine ratio). Treatment of RUPP rats with rosiglitazone, a PPARγ agonist, has been shown to ameliorate hypertension, improve vasorelaxation, and reduce ACR [89]. Interestingly, this beneficial effect is blocked by an HO-1 (heme oxygenase-1) inhibitor, SnPP [89], suggesting that the regulatory role of PPARγ may be dependent on the HO-1 pathway.

On the other hand, treatment with T0070907, an antagonist of PPARγ, in mice has been shown to induce hallmark symptoms of PE, including hypertension, proteinuria, and fetal growth restriction [90,91]. These mice also displayed increased total placental sFlt-1, increased HO-1, and decreased VEGF, as well as decreased overall labyrinth trophoblast differentiation, which are parameters associated with PE [77].

Both animal models and human tissue studies have shown a decrease in PPARγ expression in the pathogenesis of induced or diagnosed PE, respectively. This suggests that PPARγ could be targeted by pharmacological interventions to potentially reduce the severity of PE in pregnant women diagnosed with the disease.

### 5.3. Potential Treatments of Preeclampsia Targeting PPARγ

Potential treatments for PE can target PPARγ or genes associated with PPARγ, leading to significant changes in PPARγ expression that positively impact women or result in mice displaying hallmark signs of the disease. Treatment with the PPARγ antagonist T0070907 induced PE-like symptoms in mice [91]. However, administration of aspirin reversed T0070907-induced changes in VEGF, sFlt, and MMP2 in both maternal blood and placental tissue, and increased the expression of PPARγ by inhibiting the cyclooxygenase (COX) pathway [90]. Interestingly, different doses of aspirin showed varying impacts on modulating PE, with a higher dose (20 mg/kg) exhibiting a more significant improvement in maternal blood pressure compared with a lower dose (10 mg/kg) [92]. Angiotensin, a vasodilator hypothesized to inhibit the COX-2 pathway [93], was also studied for its effects on PE. In a study using a rat model of PE (RUPP rat), treatment with angiotensin 1–7 elevated the expression of PPARγ, leading to a significant decrease in systolic blood pressure and other PE symptoms. The researchers speculated that angiotensin 1–7 may increase PPARγ expression by enhancing the actions of the endothelial nitric oxide synthase (eNOS) [94]. Overall, PPARγ appears to be a promising candidate for early diagnostic biomarkers and treatment targets for PE, but further studies are needed to elucidate the underlying mechanisms.

## 6. PPARγ Functions in Placentas from GDM

Gestational diabetes mellitus (GDM), a form of glucose intolerance that arises during pregnancy, affects approximately 7.6% of pregnancies in the US and is one of the most common obstetric complications [95]. According to a 2020 report by BlueCross BlueShield, the incidence of GDM has increased by 16.6% from 2014 to 2018 [1]. GDM can result in short-term and long-term complications for both the mother and the fetus. Short-term effects include fetal macrosomia (large birth weight), hypoglycemia, respiratory distress syndrome, and preterm birth. Later in life, both the mother and the child are at increased risk of developing Type 2 Diabetes Mellitus (T2DM) [96]. Additionally, GDM pregnancies can also lead to high-risk pregnancy complications such as PE and miscarriage [97,98]. Alarmingly, at least 20% of GDM pregnancies were reported to develop pregnancy-induced hypertension [8]. However, the diagnosis of GDM is challenging due to discrepant diagnosis criteria and a lack of noticeable symptoms [99].

As the intermediate transportation site between mother and fetus, placentas from GDM also displayed a pathophysiological change in the spiral artery and vasculature [100]. While the placenta does not require insulin as a glucose regulator since glucose is the primary energy source for both the placenta and the fetus, the placenta still expresses insulin receptors, making it sensitive to maternal hyperglycemia. This sensitivity positively correlates with fetal growth and macrosomia [101]. Currently, insulin, metformin, and insulin detemir are the only FDA-approved drugs for the treatment of GDM [102]. Interestingly, most gene expression alterations in GDM occur in the lipid pathway rather than the glucose pathway, and are mostly associated with dyslipidemia and insulin resistance [15]. Dysregulation of PPARγ, which is involved in fatty acid storage, glucose storage, and insulin sensitivity, has been implicated in GDM. Omega-3 supplements for women with GDM have been shown to decrease PPARγ expression and fasting blood glucose levels while increasing PPARγ expression in peripheral blood mononuclear cells (PBMCs) [103].

PPARγ agonists, such as rosiglitazone, are commonly used drugs to treat T2DM [14]. However, due to the inability of rosiglitazone to cross the placenta during early gestation [104] but potential transfer and metabolism by embryos during late gestation [105,106], studies have been conducted to assess its safety during pregnancy. Although limited data support its safety, several studies have shown that low doses of rosiglitazone do not have adverse effects on fetal development [107,108,109,110]. However, some studies have demonstrated that activation of PPARγ by rosiglitazone disrupts the vascularity and morphology of murine placenta [111,112]. Meanwhile, only limited research demonstrated its benefit in fetal development. One study showed that rosiglitazone ameliorated the adverse effects caused by nicotine exposure including abnormal cell death and suppressed angiogenesis in murine conceptus [113]. Therefore, it is crucial to further explore and understand the impact of PPARγ and its ligands in GDM, particularly in the placenta.

### PPARγ Function in the Placentas of GDM Patients

The expression changes of PPARγ in the placentas of patients with GDM have yielded inconsistent results across various studies. While some studies have reported downregulation of placental PPARγ, This can result in placental ischemia, or reduced blood flow to the placenta, due to failed remodeling of spiral arteries and impaired vascularization expression in GDM patients compared with healthy control groups [76,114,115], particularly in syncytiotrophoblasts and EVT [116]; other studies have found increased gene expression of PPARγ in the placentas of GDM patients, including in Australian women with GDM [117] and in the trophoblast choriocarcinoma BeWo cell line under hyperglycemic conditions [118], which coincided with suppressed cell proliferation. Interestingly, PPARγ was also found to be upregulated in PBMCs of women with GDM [103], and leukocyte PPARγ mRNA levels were significantly higher in GDM patients compared with those of healthy patients [119].

In vitro studies have demonstrated that hyperglycemic conditions can impair placental vascularity through the repression of migration and viability [120]. PPARγ has been studied as a pharmaceutical target for T2DM prevention or treatment in GDM, and several pathways have been implicated in the PPARγ-mediated regulation of GDM from various aspects. For example, exogenous activation of PPARγ by 15dPGJ2 has been shown to prevent nitric oxide overproduction in the placenta of pre-gestational diabetic women [121]. Another study found that C1q/tumor necrosis factor-related protein 6 (CTRP6) interacts with PPARγ to regulate trophoblast function, with both CTRP6 and PPARγ being upregulated in high glucose-induced HTR-8/SVneo cells [122]. Depletion of CTRP6 rescued viability, invasion, and migration in HTR8/SVneo cells, while PPARγ overexpression blocked the protective effects of CTRP6 downregulation [122]. Additionally, a novel adipokine regulated by PPARγ in trophoblasts was discovered, with the administration of PPARγ agonists rosiglitazone and GW1929 elevating the expression of Chemerin and activation of the AKT/PI3K pathway [123]. Depletion of the chemerin receptor chemokine-like receptor 1 restrained this effect. Disulfide-bond A oxidoreductase-like protein (DsbA-L), an enzyme that regulates fat deposition, was also identified as a downstream target of PPARγ [124]. Rosiglitazone was found to improve insulin sensitivity through interaction with DsbA-L in the HTR-8/Svneo cell line, with upregulation of DsbA-L being crucial for the function of rosiglitazone in the PI3K-PKB/AKT pathway [125].

## 7. Future Directions and Conclusions

Currently, there is substantial evidence supporting the critical role of PPARγ in placental biology. PPARγ is highly expressed in trophoblasts, and its expression is influenced by maternal factors. While numerous in vitro studies have demonstrated that PPARγ regulates trophoblast differentiation and migration, there is still limited in vivo information that provides a comprehensive understanding of placental physiology and the complex pathophysiology of pregnancy. Several genes and chemicals have been identified to interact with PPARγ in trophoblasts, including COX, 11β-HSD2, angiotensin 1–7, HO-1, GCM1, Nrf2, ANGPTL4, CTRP6, DsbA-L, and aspirin. These interactions offer potential mechanistic insights into the function of PPARγ in the placenta. However, the current conclusion of PPARγ in placentas affected by GDM remains inconsistent. Determining whether PPARγ plays a protective role in the progression of GDM requires further in vivo studies, as systematic hyperglycemia may differ from that observed in in vitro models.

Furthermore, the ability of PPARγ ligands to modulate trophoblast function underscores the promising potential of PPARγ as a therapeutic target in the treatment of conditions. However, as depicted in Figure 2, apart from the extensively studied rosiglitazone, the effects of other ligands on preeclampsia require additional information, as each ligand may have distinct actions. Studies regarding the safety and effect of PPARγ ligands on treating GDM still need future attention. Continued research in this field holds great promise for advancing our understanding of PPARγ function in the placenta and developing innovative therapeutic strategies for managing complicated pregnancies.

## Figures and Tables

**Figure 1 nutrients-15-02459-f001:**
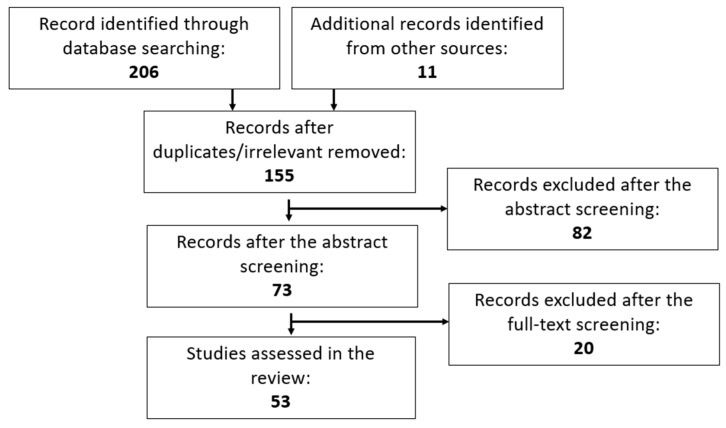
PRISMA Flowchart of article selection process.

**Figure 2 nutrients-15-02459-f002:**
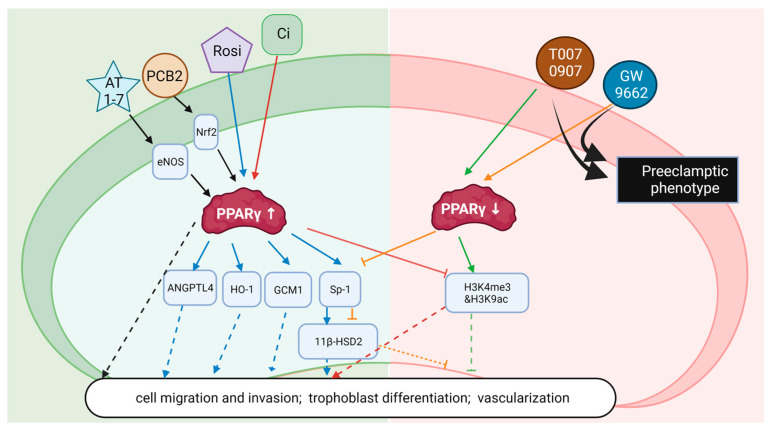
Illustrative representation of ligands and genes that interact with PPARγ in regulating placenta function related to preeclampsia. PPARγ activated by agonists or upstream regulation by Angiotensin 1-7 or Procyanidin B2 ameliorate preeclamptic phenotypes. Decreased PPARγ activity triggered by antagonists exacerbates the preeclamptic condition. AT1-7: Angiotensin 1-7; PCB2: Procyanidin B2; Rosi: Rosiglitazone: Ci: Ciglitazone.

**Table 1 nutrients-15-02459-t001:** Natural and synthetic ligands of PPARγ *.

Agonists		Antagonists	
Natural Ligand	Synthetic Ligand	Natural Ligand	Synthetic Ligand
Unsaturated fatty acid ** [15]	GW1929 [16]	Betulinic acid [17]	SR-202 [18]
Oxidized LDL [19]	TZD ***	NFκB [20]	BADGE [21]
EETs [22]	FMOC-L-Leucine [23]	Fetuin A [20]	LG100641 [24]
15d-PGJ2 [25]	INT131 [26]		PD068235 [27]
Azelaoyl phosphatidylcholine [28]	Farglitazar (GI262570) [29]		T0070907 [30]
9-oxoODE [31]	S26948 [32]		GW9662 [33]
13-oxoODE [34]	AZ 242 [35]		
15-HETE [36]	LG100754 [37]		
13-HODE [36]			

* Partial ligands of PPARγ such as telmisartan [38], Irbesartan [39], metaglidasen [40], and non-TZD partial agonist (nTZDpa) [41] are not included. ** poly-unsaturated FAs γ-linolenic (18:3), eicosatrienoic acid (C20:3), dihomo-γ-linolenic (20:3), arachidonic acid (C20:4), and eicosapentaenoic acid (C20:5). *** rosiglitazone, pioglitazone, troglitazone, ciglitazone [21], RWJ-241947 [42], NC-2100 [43], and KRP-297 [44].

## Data Availability

Data available in a publicly accessible repository.

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
