# Peer review of "Understanding PPARγ and Its Agonists on Trophoblast Differentiation and Invasion: Potential Therapeutic Targets for Gestational Diabetes Mellitus and Preeclampsia"

_nutrients, 2023, doi:10.3390/nu15112459_

Round 1

Reviewer 1 Report

In general, the manuscript is well written and organized. It also provides valuable and useful summary of the subject. Major issues are lack of recent studies and lack of critical point of view. While most of the issues and conflicting results are well identified, there is a lack of possible explanations, possible suggestions for further studies, which would greatly increase the scientific contribution of this manuscript.

Introduction - The primary focus of this paper, as implicated in the Title, is PPARγ role in GDM. The introduction would benefit from expanding it with few sentences regarding the role of PPARγ in insulin resistance, glucose metabolism, inflammation, while currently only short information about its lipid metabolism regulator properties is mentioned.

Line 30 – delete the “an” in “an additional .. acids”

Line 78 – please delete the phrase “context-dependent”, is there a possible difference in study designs which could explain conflicting results?

Line 95 – delete “and context”, or define which “contexts” you refer to, differences in study designs, cell types as mentioned or something else?

Line 98 – diabete = diabetes

Line 102 – add rs201018 “polymorphism”

Line 118 – consider renaming this caption “PPARγ functions in the placentas from preeclampsia”, such as “The role of PPARγ in preeclampsia” 

Line 128 – delete “as well as an increased risk of cardiovascular disease in their children”, just as well as their children is enough

Line 138 – replace failed = inadequate, aberrant..

Line 185 – is there a possible explanation for these conflicting results or suggested model for studying this? If there is, please provide in this section.

Line 239 – this line is simply wrong, insulin cannot be “a source of energy”, please paraphrase this sentence, (regulator, etc.)

Line 251-256 – No adverse effects, however, are there positive effects for the fetus – such as lower incidence for further complications? If so please provide information and literature. Define in vivo, murine models?

Conclusion – “Importantly, the ability of PPARγ ligands, particularly rosiglitazone, to modulate trophoblast function underscores the promising potential of PPARγ as a therapeutic target in the treatment of conditions such as PE and GDM.” This is oversimplification of PPARγ ligands role, please mention the limitations and issues you also discussed in the GDM section.

References – This paper critically lacks up-to-date studies, only few recent studies are mentioned, however, many have been published.

Quality of English Language is good.

Author Response

Introduction and main text: We thank the reviewer’s comments. Per suggestion, we edited the language and corrected the typos. Please see lines 66, 117, 134, 138, 142, 161, 172, 180, 226, 288, 306. We also added information regarding other roles of PPARγ. Please see lines 35 &81.

Conclusion: We thank the reviewer’s insightful comments. We added the limitations and issues of PPARg ligands and further discussed the gaps regarding PPARg ligands.

Reference: We thank the reviewer’s insightful comments to improve the overall quality of the manuscript. We have added more recent studies. There are still relatively limited recent studies regarding PPARg in GDM compared to PE and we discussed this in the main text. 

Reviewer 2 Report

The authors have provided a review of the current understanding of the role of PPARγ in trophoblast differentiation and invasion during pregnancy and how its dysfunction may contribute to the development of gestational diabetes mellitus (GDM) and preeclampsia (PE). The authors also summarize the potential of PPARγ agonists, particularly rosiglitazone, in modulating trophoblast function and as a therapeutic target for GDM and PE.

1. The authors have provided a relatively comprehensive summary of the relevant literature on the topic. However, it would be beneficial to include more recently published reports in addition to the predominantly older articles that are currently referenced.

2. Given the recent publication of other papers that examine the role of Peroxisome Proliferator-Activated Receptors (PPARs) in trophoblast physiology (e.g., Int J Mol Sci. 2021 Jan; 22(1): 433.), there are concerns around the novelty of the review

3. While the authors have provided a thorough summary of the role of PPARγ in trophoblast differentiation and invasion, there are no further comments or analyses presented regarding potential gaps in the current research or how the field may evolve in the future

4. There are no visually informative figures to illustrate the mechanisms described in the review, which could be added

Overall the language is good and clear

Author Response

  1. We thank the reviewer’s insightful comments to improve the overall quality of the manuscript. We have added more recent studies. There are still relatively limited recent studies regarding PPARg in GDM compared to PE and we discussed this in the line 355-358.
  2. We thank the reviewer’s insightful comments. We have added more studies.
  3. We agree with the reviewer. We have added the discussion regarding the gaps and future direction to improve the overall quality.
  4. We thank the reviewer’s insightful comments. A graphic image has been added to illustrate the mechanisms. 

Reviewer 3 Report

The review submitted by Qin Y et al., discusses the role of PPAR-gamma in trophoblast differentiation and invasion, and its mechanistic relevance to pregnancy complications such as gestational diabetes and preeclampsia. Although interesting, the review fails to summarize or cite the published literature. For instance, a very well-written review published by Psilopatis I on “The role of peroxisome proliferator-activated receptors in preeclampsia” is not even cited. The authors should include the article selection process and the search terms used to identify relevant papers.      

Minor editing of the English language is required. 

Author Response

We thank the reviewer’s insightful comments to improve the overall quality of the manuscript. We have added more recent studies.

We agree with the reviewer. The article selection process with flow chart and the search terms used were added to the manuscript. 

Round 2

Reviewer 1 Report

The revision improved manuscript significantly.

Reviewer 3 Report

The authors have addressed the comments appropriately.